# Changes in Smoking and Vaping over 18 Months among Smokers and Recent Ex-Smokers: Longitudinal Findings from the 2016 and 2018 ITC Four Country Smoking and Vaping Surveys

**DOI:** 10.3390/ijerph17197084

**Published:** 2020-09-27

**Authors:** Shannon Gravely, Gang Meng, K. Michael Cummings, Andrew Hyland, Ron Borland, David Hammond, Richard J. O’Connor, Maciej L. Goniewicz, Karin A. Kasza, Ann McNeill, Mary E. Thompson, Sara C. Hitchman, David T. Levy, James F. Thrasher, Anne C.K. Quah, Nadia Martin, Janine Ouimet, Christian Boudreau, Geoffrey T. Fong

**Affiliations:** 1Department of Psychology, University of Waterloo, Waterloo, ON N2L 3G1, Canada; shannon.gravely@uwaterloo.ca (S.G.); gmeng@uwaterloo.ca (G.M.); ackquah@uwaterloo.ca (A.C.K.Q.); nadia.martin@uwaterloo.ca (N.M.); j2ouimet@uwaterloo.ca (J.O.); 2Department of Psychiatry & Behavioral Sciences, Medical University of South Carolina, Charleston, SC 29425, USA; cummingk@musc.edu; 3Department of Health Behavior, Roswell Park Comprehensive Cancer Center, Buffalo, NY 14203, USA; Andrew.Hyland@RoswellPark.org (A.H.); Maciej.Goniewicz@RoswellPark.org (M.L.G.); karin.kasza@roswellpark.org (K.A.K.); 4School of Psychological Sciences, University of Melbourne, Melbourne, VIC 3010, Australia; rborland@unimelb.edu.au; 5School of Public Health and Health Systems, University of Waterloo, Waterloo, ON N2L 3G1, Canada; david.hammond@uwaterloo.ca; 6Addictions Department, Institute of Psychiatry, Psychology & Neuroscience, King’s College London, London WC2R 2LS, UK; ann.mcneill@kcl.ac.uk (A.M.); sara.hitchman@kcl.ac.uk (S.C.H.); 7Shaping public health policies to reduce inequalities & harm (SPECTRUM), The University of Edinburgh, Edinburgh, EH8 9YL, UK; 8Department of Statistics and Actuarial Science, University of Waterloo, ON N2L 3G1, Canada; methompson@uwaterloo.ca (M.E.T.); cboudreau@uwaterloo.ca (C.B.); 9Lombardi Comprehensive Cancer Center, Georgetown University, Washington, DC 20057, USA; dl777@georgetown.edu; 10Department of Health Promotion, Education & Behavior, Arnold School of Public Health, University of South Carolina, Columbia, SC 29208, USA; THRASHER@mailbox.sc.edu; 11Tobacco Research Department, Center for Population Health Research, National Institute of Public Health, Cuernavaca 62100, Mexico; 12Ontario Institute for Cancer Research, Toronto, ON M5G 0A3, Canada

**Keywords:** smoking, transitions, tobacco, nicotine vaping products, adults, e-cigarettes

## Abstract

This descriptive study of smokers (smoked at least monthly) and recent ex-smokers (quit for ≤2 years) examined transitions over an 18 month period in their smoking and vaping behaviors. Data are from Waves 1 (W1: 2016) and 2 (W2: 2018) of the ITC Four Country Smoking and Vaping Survey, a cohort study of adult (≥18+) smokers, concurrent users (smoke and vape), and recent ex-smokers from Australia, Canada, England, and the United States (US). Respondents (N = 5016) were classified according to their smoking and vaping status, which resulted in eight subgroups: (1) exclusive daily smokers (2) exclusive non-daily smokers; (3–6) concurrent users (subdivided into four groups by each combination of daily/non-daily smoking and daily/non-daily vaping); (7) ex-smokers who vape; (8) ex-smokers not vaping. The analyses focused first on describing changes between groups from W1 to W2. Second, transition outcomes were assessed based on changes in smoking and vaping between W1 and W2. Transitions focused on smoking were: no change in smoking (continued smoking at the same frequency); decreased smoking; increased smoking; discontinued smoking; relapsed (ex-smokers at W1 who were smoking at W2). Transitions focused on vaping were: initiated vaping; switched from smoking to vaping. Overall, this study found that the vast majority of smokers were smoking 18 months later. Non-daily smokers were more likely than daily smokers to have discontinued smoking (*p* < 0.0001) and to have switched to exclusive vaping (*p* = 0.034). Exclusive non-daily smokers were more likely than exclusive daily smokers to have initiated vaping (*p* = 0.04). Among all W1 daily smokers, there were no differences in discontinued smoking between daily smokers who vaped (concurrent users) and exclusive daily smokers; however, concurrent users were more likely than exclusive daily smokers to have decreased to non-daily smoking (*p* < 0.001) or to have switched to vaping by W2 (*p* < 0.001). Among all W1 non-daily smokers, there were no significant differences in increased smoking or discontinued smoking between concurrent users or exclusive smokers. Most ex-smokers remained abstinent from smoking, and there was no difference in relapse back to smoking between those who vaped and those who did not.

## 1. Introduction

Cigarettes remain the most dangerous and commonly consumed nicotine product [1]. However, an increasing number of smokers are also using non-combusted tobacco/nicotine products, such as e-cigarettes, heated tobacco products, and snus [1,2,3]. The most popular and rapidly growing class of these non-combustible products are nicotine vaping products (NVPs, commonly known as e-cigarettes) [3]. Evidence suggests that completely substituting NVPs for combustible cigarettes greatly reduces users’ exposure to numerous toxicants and carcinogens [2,4,5].

Over the last decade, scientists, clinicians, advocates, and public health organizations have debated whether or not NVPs yield a net benefit to population health [2]. The key question with respect to smokers, is whether NVPs can help them transition away from cigarettes, and remain abstinent from smoking. A recent randomized trial found that among a sample of 886 adult smokers seeking help to quit smoking, significantly more smokers using an NVP quit smoking after one year (18.0%) compared to smokers receiving nicotine replacement therapy (NRT) (9.9%) [6]. Another randomized trial found that combination therapy of nicotine patches with a nicotine e-cigarette was associated with a modest improvement in continuous abstinence at six months (7%) compared to NRT plus a nicotine-free e-cigarette (4%) or NRT alone (2%) [7]. However, findings from randomized trials may not be generalizable to whether NVPs are associated with reducing or discontinuing smoking in the manner in which they are used by smokers in the natural environment. It is important to understand the extent to which vaping can facilitate transitioning away from smoking in “real-world” settings, even as we recognize the challenges in making causal inferences with non- randomized trial study designs.

The increasing diversification of the NVP landscape has implications for understanding how product use changes over time (e.g., transitioning between combustible and non-combustible products), and longitudinal cohort studies are essential for assessing these transitions [8,9]. Examining patterns of smoking and vaping, especially exploring the nature of these transitions between cigarettes and NVPs over time, is a beginning point for describing transitioning in and out of smoking, and how vaping may play a role in these transitions.

Using the Population Assessment of Tobacco and Health (PATH) Study (a national population longitudinal study that tracks tobacco product use in a representative sample of adults in the United States (US)), Hyland et al. (2020) described patterns of tobacco/nicotine product use among current and former smokers across time [9]. The main categories included: persistent use (continued use across multiple time points), discontinued use (use to no use), relapse (stopped use and began use again), switching (changing between product types), and inconsistent use (back and forth between use and non-use). Evidence from the PATH Study has shown that the majority of cigarette smokers persist with smoking across time [10,11], whereas transition behaviors among smokers initiating and persisting with NVP use (concurrent users) are more highly variable [11,12,13]. Moreover, continuing with/transitioning toward cigarettes, is more common than continuing with/transitioning toward NVPs. When it comes to differential transitions away from smoking by vaping status, Coleman et al. [12] found that among baseline concurrent users, daily vapers were more likely than non-daily vapers to report smoking abstinence at follow-up. Kasza et al. [13] found that vaping was positively associated with making attempts to quit, but was not associated with discontinuing smoking among attempters. However, some caution is required when interpreting population-based studies as NVP use is not randomized and there is reason to believe those vaping differ from those who are not. For example, concurrent users have been shown to be more dependent on nicotine than smokers who do not vape [14,15]. Further, baseline smokers who use NVPs could be viewed as “treatment failures” if they initiated NVP use to help them to stop smoking, but had not quit at the time of the follow-up survey.

In this current prospective cohort study, data from the ITC Four Country and Vaping Surveys (ITC 4CV) were used to explore and describe behavioral transitions among smokers and recent ex-smokers in “real world” conditions at two time points (18 months apart) in four high-income countries, allowing us to examine transitions between smoking and vaping across a broader international context. Specifically, this study examined point prevalence states of smoking and vaping at baseline and follow-up among exclusive smokers (daily and non-daily/past-daily; no vaping), concurrent users (based on both smoking and vaping frequency), and recent ex-smokers (who either vaped or not). Exploratory analyses were also conducted to compare transition outcomes at follow-up between subgroups (based on baseline vaping or not): no change; decreased smoking (from daily to non-daily); increased smoking (from non-daily to daily); discontinued smoking; switched from smoking to vaping; initiated vaping. Relapse back to smoking was assessed for ex-smokers based on baseline vaping or not. Finally, we conducted difference-in-differences (DID) analyses by country (as NVP policies vary across these four countries), to test whether smokers who were vaping were more likely to decrease or discontinue smoking (compared to those who were not vaping). The authors did not have any preconceived hypotheses, as these exploratory analyses were based on findings from the transition estimates.

## 2. Methods

### 2.1. Study Sample and Procedure

The ITC 4CV Survey is a longitudinal cohort study that consists of four parallel online surveys conducted in Canada, the US, England, and Australia. In addition to respondents retained from the ITC Four Country Survey (the predecessor of ITC 4CV), adults (≥18 years) were recruited by commercial panel firms in each country at Wave 1 (W1: July-November 2016) as cigarette smokers (smoked at least 100 cigarettes in their lifetime and smoked at least monthly), recent ex-smokers (quit within ≤2 years), or at-least-weekly NVP users (vapers). The sample in each country was designed to be as representative as possible of cigarette smokers, ex-smokers, and NVP users (e.g., by age, sex, and region). All W1 respondents were invited back to complete the Wave 2 (W2: February-July 2018) survey, and were included in the current study if they were at least monthly smokers or recent ex-smokers at W1 and completed W2. All those who were non-daily smokers at baseline must have smoked daily in the past to be considered for inclusion. All respondents provided consent, and all study procedures were approved by relevant research ethics boards. Further details about the ITC 4CV study can be found in the 2016 [16] and 2018 [17] technical reports, and in Thompson et al. [18].

### 2.2. Measures

The surveys, with original response options, can be found at the ITC Project website: https://itcproject.org/surveys/. The following variables were used in the current study:

#### 2.2.1. Wave 1 (2016) Baseline Measures

Sociodemographic variables: sociodemographic data were collected by the commercial panels and verified at the time of survey completion, including: age, gender, education, and country of residence.

Smoking and vaping status: respondents who reported smoking at least monthly (daily vs. non-daily) or recently quit smoking at baseline were included in this study. A parallel question was asked about NVP use and vaping frequency, and users were categorized as: current daily vapers, current non-daily vapers, or not currently vaping (those who were vaping less than monthly were considered non-vapers).

Time-in-sample (TIS): the analyses controlled for the time-in-sample (TIS), the number of waves that the respondent completed. TIS has been found to be related to differences in responses to a number of outcome variables in past ITC studies. Methodological details of TIS are presented elsewhere [19].

#### 2.2.2. Country NVP Regulations

Country regulations covering NVPs have been previously summarized [20]. In brief, in 2016, the US allowed NVPs to be regulated and sold with few federal restrictions, England allowed NVPs to be sold and were regulated under the UK Tobacco and Related Products Regulations [21]. In contrast, NVPs were not permitted to be legally sold in Canada (with weak enforcement) or Australia (with strong enforcement). In 2018, the only major change in national regulatory policies was that the Canadian government passed the Tobacco and Vaping Products Act [22], which allowed NVPs to be regulated and sold as of May 2018.

#### 2.2.3. Classification of Point Prevalence Smoking and Vaping Status

The respondents were categorized in terms of daily and non-daily smoking and vaping as described in Table 1. The categorization of concurrent users is according to the system developed by Borland et al. [23].

#### 2.2.4. Wave 2 (2018) Transition Outcomes

Transition outcomes were determined by respondents’ self-reported smoking and vaping statuses at W1 and W2. The following seven transition outcomes were deemed relevant to the research question, and were adapted from the approach described by Hyland et al. [9]. The transition outcomes focused only on smoking (A–E) were: (A) no change in smoking status (continued smoking at the same frequency), (B) decreased smoking, (C) increased smoking, (D) discontinued smoking, and (E) relapsed (ex-smokers at W1 who were smoking at W2). Transition outcomes focused on vaping (F and G) were: (F) initiated vaping; (G) switched from smoking to vaping (a sub-analysis of (4) above).

### 2.3. Statistical Analyses

Unweighted frequencies were used to describe the respondents’ baseline (W1) characteristics (Table 2). All subsequent analyses were weighted using longitudinal weights that were computed for all respondents. In brief, a raking algorithm [24] was used to calibrate the weights on smoking status, geographic region, and demographic measures (e.g., sex, age, ethnicity, and education). This calibration was done using benchmarks from national surveys from each of the respective countries. Generalized estimating equations (GEEs) using the predicted marginal standardization method [25] were used to generate regression models (PREDMARG) to estimate the point prevalence states of smoking and vaping at both time points.

There were three main sets of analyses conducted for this study. First, multivariable logistic regression models were used to descriptively examine within-person transitions between states of smoking and vaping (based on Table 1 group classifications: 1–8) among W1 daily smokers (*n* = 3983; exclusive daily smokers, predominant smokers, dual-daily users), non-daily smokers (*n* = 453; exclusive non-daily smokers, concurrent non-daily users, predominant vapers); recent ex-smokers (*n* = 580; vaping or not vaping) (Table 3). These descriptive analyses focused on changes between groups that occurred in W2 (as well as remaining in the same state of smoking and/or vaping) and estimated point prevalence W2 estimates for the applicable transitions. The analyses for W1 daily smokers and recent ex-smokers controlled for age, country, education, and TIS (gender was not significant in the bivariate analyses and therefore was not used as a covariate in the larger model). Given the relatively small sample sizes, the analysis for W1 non-daily smokers used the same covariates as the previous model, with the exception of country. Additionally, groups 5 and 6 (predominant vapers and concurrent non-daily users) were combined at W2 due to small sample sizes.

The second set of analyses were exploratory in nature and compared outcomes between subgroups based on the seven transitions outlined in Section 2.2.4. The transition outcomes focused on smoking were: (A) no change (continued smoking at the same frequency)—(i) exclusive daily smokers vs. daily smokers who were vaping; (ii) exclusive non-daily smokers vs. non-daily smokers who were vaping; (B) decreased smoking from daily to non-daily—exclusive daily smokers vs. daily smokers who were vaping; (C) increased smoking—exclusive non-daily smokers vs. non-daily smokers who were vaping; (D) discontinued smoking—(i) all daily smokers vs. all non-daily smokers; (ii) exclusive daily smokers vs. daily smokers who were vaping; (iii) exclusive non-daily smokers vs. non-daily smokers who were vaping; (iv) between the concurrent user groups; (E) relapsed back to smoking—ex-smokers who were vaping vs. ex-smokers who were not vaping. Transition outcomes focused on vaping were: (F) initiated vaping—exclusive daily smokers vs. exclusive non-daily smokers; (G) discontinued smoking and switched to vaping—(i) all daily smokers vs. all non-daily smokers; (ii) exclusive daily smokers vs. daily smokers who were vaping; (iii) exclusive non-daily smokers vs. non-daily smokers who were vaping; (iv) between the concurrent user groups (Table 4). The same covariates were used in the models as described above.

The third set of analyses tested differences by country and baseline NVP use (yes or no) for: (1) daily smokers who decreased to non-daily smoking or discontinued smoking between W1 and W2; (2) non-daily smokers discontinuing smoking between W1 and W2. These analyses examined if there were differences within countries, and if these differences differed between countries (using the DID method) [26]. The analyses controlled for age, education, TIS, and uptake of NVP use between W1 and W2 (regardless of vaping history).

Statistical significance and confidence intervals were computed at the 95% confidence level. Analyses were conducted in SAS-Callable SUDAAN (V.11; RTI International, Research Triangle Park, NC, USA).

### 2.4. Ethics approval

Study questionnaires and materials were reviewed and provided clearance by Research Ethics Committees at the following institutions: University of Waterloo (Canada, ORE#20803/30570, ORE#21609/30878), King’s College London, UK (RESCM-17/18-2240), Cancer Council Victoria, Australia (HREC1603), University of Queensland, Australia (2016000330/HREC1603); and Medical University of South Carolina (waived due to minimal risk).

## 3. Results

Overall, 5632 W1 respondents in the larger cohort study were followed up and had complete data at W2. Those who had never smoked (*n* = 24), smoked less than monthly (*n* = 255), were never daily smokers (*n* = 136), or quit smoking more than 2 years ago (*n* = 201) were excluded for this study, thus resulting in 5016 respondents being included in the analyses: exclusive (at least monthly) smokers (*n* = 3319), concurrent users (concurrently smoke and vape at least monthly, *n* = 1117), and recent ex-smokers (*n* = 580: of whom 33.5% quit smoking within the last 6 months, 25.0% between 7 and 12 months, and 41.5% between 1 and 2 years ago). Respondent baseline characteristics are presented in Table 2.

### 3.1. User Group Transitions among Daily Smokers and Non-Daily Smokers: Point Prevalence W2 Estimates

Table 3 shows the transitions between the eight subgroups between W1 and W2. Transitions are briefly described below:

### 3.2. Transitions among Daily Smokers

W1 exclusive daily smokers: At W2: 71.4% did not change (remained exclusive daily smokers and not vaping), 3.0% decreased to non-daily smoking, 14.1% became concurrent users, and 11.6% were ex-smokers (discontinued smoking).

W1 predominant smokers: At W2: 25.4% did not change (remained smoking daily and vaping less than daily), 42.2% became exclusive daily smokers, 2.7% became exclusive non-daily smokers, 9.5% were ex-smokers (discontinued smoking).

W1 dual-daily users: At W2: 38.5% did not change (remained using both products daily), 21.4% became exclusive daily smokers, 1.5% became exclusive non-daily smokers, and 11.7% were ex-smokers (discontinued smoking).

### 3.3. Transitions among Non-Daily Smokers

W1 exclusive non-daily smokers: At W2: 31.2% did not change (remained exclusive non-daily smokers), 20.6% became concurrent users, 21.1% became exclusive daily smokers, and 27.3% were ex-smokers (discontinued smoking).

W1 predominant vapers: At W2: 39.7% did not change (remained vaping daily and smoking non-daily), 4.2% became exclusive daily smokers, 1.7% became exclusive non-daily smokers, and 23.8% were ex-smokers (discontinued smoking).

W1 concurrent non-daily users: At W2: 51.2% did not change (remained using both products on a non-daily basis), 5.3% became exclusive daily smokers, 9.9% became exclusive non-daily smokers, and 17.6% were ex-smokers (discontinued smoking).

### 3.4. Transitions among Recent Ex-Smokers

W1 recent ex-smokers who vaped: 67.3% did not change (were still exclusively vaping and not smoking), 4.4% were exclusively smoking, and 8.1% became concurrent users.

W1 recent ex-smokers who did not vape: 82.1% did not change (remained abstinent from smoking and vaping), 11.3% were exclusively smoking, and 2.0% became concurrent users.

### 3.5. Wave 1 to Wave 2 Transition Subgroup Comparisons

Table 4 presents the subgroup comparisons (including the transition code) for each of the seven transition outcomes. The main subgroup comparisons are outlined below.

### 3.6. Comparisons between Wave 1 Daily and Non-Daily Smokers

Non-daily smokers were more likely than daily smokers to have discontinued smoking (transition code: D(i)) or to have switched to vaping (G(i)). Exclusive non-daily smokers were more likely than exclusive daily smokers to have initiated vaping between W1 and W2 (F).

### 3.7. Comparisons between Wave 1 Daily Smokers: Exclusive Daily Smokers vs. Concurrent Users

Decreasing smoking differed between exclusive smokers and concurrent users, where predominant smokers and dual-daily users were significantly more likely than exclusive daily smokers to have reduced to non-daily smoking by W2 (A(i)). There were no differences between exclusive daily smokers and concurrent users in discontinuing smoking (D(ii)). Among W1 daily smokers, concurrent users were more likely than exclusive smokers to have discontinued smoking by switching to vaping (G(ii)).

### 3.8. Comparisons between Wave 1 Non-daily Smokers: Exclusive Non-daily Smokers vs. Concurrent Users

There were no significant differences in increased smoking (C) or discontinued smoking (D(iii) between exclusive non-daily smokers and concurrent users.

### 3.9. Comparisons between Wave 1 Concurrent Users

Overall, 59.4% of W1 concurrent users were still concurrently using both products at W2 (data not shown in tables). With regard to subgroup comparisons (dual-daily users were used as the control group: the group with the most frequent smoking and vaping frequency), predominant vapers were more likely than dual-daily users to have discontinued smoking at W2 (D(iv)), but there was no statistical difference between these groups for having switched to vaping (G(iv)), likely owing to the small sample size of predominant vapers (resulting in a large 95% confidence interval). There were no other differences.

### 3.10. Comparisons between Recent Ex-smokers, Vapers and Non-Vapers:

There were no differences in the relapse rates back to smoking between vapers and non-vapers (E). However, significantly fewer ex-smoking vapers transitioned to exclusive smoking compared to non-vapers (*p* = 0.045, shown in Table 3).

### 3.11. Country Differences

Table 5 presents daily smokers’ and non-daily smokers’ progression away from smoking by country and vaping status. In brief, there were considerable similarities between countries, but there were some differences: (1) daily smokers who vaped were more likely to have decreased from daily to non-daily smoking than smokers who did not vape in Canada (*p* = 0.002) and England (*p* = 0.03). This was not statistically significant in the US or Australia, although trended in the same direction in the US (*p* = 0.06); (2) daily smokers who vaped were less likely than those who did not vape to discontinue smoking in Canada (*p* = 0.003). This was not found for Australia, the US or England; (3) non-daily smokers who vaped were significantly less likely to have discontinued smoking compared to those who did not vape in Australia (*p* < 0.001) (this association was not found in the other three countries); (4) cross-country analyses showed that Australia had a larger difference for discontinued smoking between vapers and non-vapers compared to Canada (*p* = 0.003) and England (*p* = 0.018).

## 4. Discussion

This study is a descriptive analysis using a representative sample of smokers and ex-smokers, and described changes in smoking and vaping over 18 months. This study has also offered some insight into whether certain smoking and vaping subgroups differed in their smoking and vaping behaviors. Overall, this study found that the vast majority of smokers were smoking 18 months later, thus reflecting the high level of stability of this behavior. We found that non-daily smokers were more likely than daily smokers to have discontinued smoking at follow-up (26.1% vs. 11.3% respectively), which is consistent with non-daily smokers being less nicotine dependent than daily smokers [15]. With regard to vaping behaviors, a lower proportion of exclusive daily smokers than exclusive non-daily smokers initiated vaping between baseline and follow-up (16.7% vs. 25.4%, respectively). Daily smokers were less likely than non-daily smokers to switch to exclusive vaping (2.9% vs. 6.5%, respectively). When comparing exclusive smokers to concurrent users, daily smokers who were vaping at baseline (concurrent use) were more likely than exclusive daily smokers to have decreased smoking (from daily to non-daily); however, we found that concurrent use at baseline was not associated with discontinued smoking for either daily or non-daily smokers. About one-third of non-daily smokers increased smoking (to daily), and there were no significant differences between non-daily smokers who vaped or did not vape. The majority of ex-smokers in this study remained abstinent from smoking (86.8%), and there were no differences in relapse between vapers and non-vapers, but baseline exclusive vapers were less likely than non-vapers to be exclusively smoking at follow-up (4.4% vs. 11.3%, respectively).

These findings suggest that smokers with established concurrent use were not more likely to discontinue smoking compared to those not vaping. This was more evident for daily smokers, who are more highly addicted to nicotine than non-daily smokers [15,27,28,29,30]. In interpreting the results, it must be noted that concurrent users are potentially more highly addicted to nicotine than exclusive smokers [23]. For example, Strong et al. [15] examined indicators of tobacco dependence across a range of tobacco products and demonstrated that concurrent users of cigarettes and NVPs had the highest mean dependence scores. This could suggest that the unassisted quit rates in such smokers may be lower than for non-vapers, so it remains possible that vaping has equalized this imbalance, rather than having no effect on cessation. The finding of more reduction is consistent with this explanation. It is also important to note that while some NVP users are vaping to quit smoking, some are vaping for other reasons which is why they sustained continued smoking [31]. In this study, among baseline smokers who also vaped, 46% planned to quit smoking within 6 months, 30% planned to quit in the future, but beyond 6 months, with the remaining 24% reporting that they did not know or did not plan on quitting, suggesting low motivation to quit smoking among many of the concurrent users. This is further supported by reasons that respondents gave for vaping, with 60% reporting that vaping may help them quit smoking, while 45% reported using an NVP for reasons other than to quit smoking (data not shown). This study, however, was not examining specific cessation attempts, but rather the naturalistic changes in nicotine product use over a period of time. Regardless, it is clear that the rates of transitioning away from smoking remain unacceptably low, and perhaps current vaping tools at best bring the likelihood of quitting up to comparable levels of less dependent smokers.

The findings of our international study are consistent with the findings of the US PATH transition studies, and other observational studies, in that most smokers remain in a persistent state of cigarette use across time, particularly the daily smokers [10,11,32,33,34]. For example, the vast majority of smokers in the PATH Study continue to smoke over time [10,11,35], and daily smoking was shown to be inversely associated with smoking abstinence [35]. Our findings are also consistent with the findings of a United Kingdom study (2016-2017) [33], in which 86% of exclusive smokers were still smoking at follow-up.

The majority of ex-smokers in our study remained abstinent from smoking, and among those who relapsed, there was no difference between vapers and non-vapers. Few other studies have examined the role of NVPs in smoking relapse. One study found that vaping may be protective against relapse [14], but similar to our study, Brose et al. [36] reported that there was no difference in relapse between exclusive daily vapers compared to those who were not vaping. However, they did find that ex-smokers who vaped infrequently had a higher probability of relapse, suggesting that this group of ex-smokers were not vaping enough to satisfy their cravings for nicotine. More research is warranted to explore how vaping may help with smoking abstinence and prevent relapse, particularly among those who may have quit for a short period of time and may be more likely to relapse compared to those who have quit for longer periods of time [37,38].

This study extends our knowledge of smoking and vaping transitions to multiple countries, and showed, for the most part, considerable similarities. However, there were some differences. For example, in Canada and England, daily smokers who also vaped were more likely to transition to non-daily smoking (compared to exclusive smokers). There was a trend in the same direction for the US and Australia; however, small sample sizes may have limited the power to detect a significant difference. Disconcertingly, among daily smokers in Canada, discontinuation of smoking was less likely for those who vaped than for those who did not vape. The estimates for England the US were slightly in the opposite direction, and were equivalent in Australia. An unexpected finding was that England did not have a higher proportion of vapers discontinuing smoking compared to the other countries considering that England has the most supportive harm reduction policies [39,40], whereas the other three countries have not taken the same approach. One unsurprising finding was that smokers from Australia (where NVPs are strictly prohibited) had very low rates of reducing smoking or discontinuing smoking among NVP users. This may be due to uncertainty of supply, or of the demands of needing to break laws to vape, acting as a disincentive to persist with vaping. Thus, the lower levels of vaping in Australia are likely related to the more restrictive laws about selling nicotine-containing e-liquids [41].

This study has several strengths. First, it is a large cohort study spanning across four countries, with differing NVP regulatory policies. Second, varying patterns in smoking and vaping frequencies over time were assessed, which is essential for characterizing daily nicotine users from non-daily nicotine users as they differ considerably (e.g., with regard to attitudes towards smoking and vaping, interest in quitting smoking, and nicotine dependence) [23]. There are, however, some limitations, therefore the findings from this paper should be interpreted with some caution. First, we only had two measurement periods, therefore we have only provided a snapshot of transition behaviors at two points in time, with no information about the intervening period (e.g., we did not explore actions taken between surveys such as use of NVPs for any quit attempts). Second, like all observational studies, vaping status was self-selected, not randomly allocated, so causal models could not be tested. Third, our sample design does not allow us to compute meaningful prevalence estimates for smoking or vaping (owing to the fact that we do not have a probability sample of the general population), thus we could not make assumptions about changes in smoking or vaping rates. Studies with a sample design involving a probability of the general population, such as the PATH Study [11], can provide prevalence estimates in addition to transitions. Fourth, some subgroups were small, thus limiting the power to detect significant differences, particularly when the sample was subdivided by country. Finally, we did not attempt to explore possible baseline differences between those in the various Wave 1 use states, so differences in transitions could be a function of any such differences rather than their baseline smoking and vaping status. We were also unable to take into account potential confounders such as tobacco/nicotine dependence, previous quit attempts, reasons for vaping, or motivation to quit smoking, mainly owing to small sample sizes in some groups. Finally, this paper is only a descriptive examination of smokers/vapers, therefore a causal interpretation of patterns is premature. Forthcoming papers will analyze the interplay between cigarettes and NVPs over time using methods that have greater potential for directly addressing possible explanations for the patterns of transitions presented in this initial descriptive study.

## 5. Conclusions

Longitudinal cohort studies are essential for assessing transitions in tobacco/nicotine product use over time, and for assessing the potential of NVPs and other nicotine delivery products for reducing the harms of smoking. Our international study confirms that the vast majority of smokers were still smoking 18 months later despite the availability of less harmful alternatives, demonstrating the persistence of cigarette smoking, the most dangerous and commonly used nicotine product. This study has also highlighted several differences between daily and non-daily smokers, particularly that non-daily smokers were more likely than daily smokers to have discontinued smoking. Notably, among daily smokers, vaping did not improve rates of discontinued smoking, but it appears to have been helpful in reducing daily smoking to non-daily smoking. Relapse rates were low among the ex-smokers in this study, and there were no differences in relapse between those who were vaping at baseline compared to those who were not vaping. Further longitudinal research is needed to examine the utility of vaping as an aid to quit smoking and if vaping can be helpful in relapse prevention. Moreover, considering the low rates of discontinued smoking in all four countries, reinforcing the need for continued public health focus on cigarette smoking prevention and cessation efforts is imperative.

## Figures and Tables

**Table 1 ijerph-17-07084-t001:** Group classification of respondents’ smoking and vaping status.

#	Group Category	Smoking	Vaping
**Exclusive smokers**			
1	Exclusive daily smokers	Daily	None
2	Exclusive non-daily smokers	Non-daily	None
**Concurrent users**			
3	Dual-daily users	Daily	Daily
4	Predominant smokers	Daily	Non-daily
5	Predominant vapers	Non-daily	Daily
6	Concurrent non-daily users	Non-daily	Non-daily
**Recent ex-smokers**			
7	Ex-smokers who vape	None	Daily/Non-daily
8	Ex-smokers not vaping	None	None

Note: Ex-smokers who vape (exclusive vapers) could not be subdivided by daily or non-daily vaping due to small sample sizes.

**Table 2 ijerph-17-07084-t002:** Respondents’ Baseline (2016) Characteristics.

Characteristics, *n* (%)		Exclusive Smokers *n* = 3319 (66.2%)	Concurrent Users *n* = 1117 (22.3%)	Ex-Smokers *n* = 580 (11.6%)	Overall N = 5016
Country	Australia	636 (19.2)	90 (8.1)	93 (16.0)	819 (16.3)
	Canada	972 (29.3)	439 (39.3)	187 (32.2)	1598 (31.9)
	England	1040 (31.3)	359 (32.1)	151 (26.0)	1550 (30.9)
	United States	671 (20.2)	229 (20.5)	149 (25.7)	1049 (20.9)
Sex	Male	1592 (48.0)	592 (53.0)	263 (45.3)	2447 (48.8)
	Female	1727 (52.0)	525 (47.0)	317 (54.7)	2569 (51.2)
Age	Mean (SD)	50.9 (13.7)	44.2 (15.2)	49.6 (14.7)	49.3 (14.5)
Age group	18–24	184 (5.5)	162 (14.5)	30 (5.2)	376 (7.5)
	25–39	536 (16.2)	298 (26.7)	126 (21.7)	960 (19.1)
	40–54	1097 (32.0)	331 (29.6)	177 (30.5)	1605 (32.0)
	55+	1502 (41.4)	326 (29.2)	247 (42.6)	2075 (41.4)
Education level	Low	1154 (34.8)	310 (27.8)	171 (29.5)	1635 (32.6)
	Moderate	1350 (40.7)	445 (39.8)	250 (43.1)	2045 (40.8)
	High	789 (23.8)	351 (31.4)	156 (26.9)	1296 (25.8)
	Not reported	26 (0.8)	11 (1.0)	3 (0.5)	40 (0.8)
Smoking Status	Daily smoking	3063 (92.3)	920 (82.4)	-	3983 (79.4)
	Non-daily smoking	256 (7.7)	197 (17.6)	-	453 (9.0)
	Recent ex-smoker	-	-	580 (100.0)	580 (11.6)
Vaping Status	Daily vaping	-	410 (36.7)	109 (18.8)	519 (10.4)
	Non-daily vaping	-	707 (63.3)	27 (4.7)	734 (14.6)
	No current vaping	3319 (100.0)	-	444 (76.6)	3763 (75.0)

Data are unweighted in order to describe the sample used in the analyses; SD: standard deviation.

**Table 3 ijerph-17-07084-t003:** User group transitions in smoking and vaping among daily smokers, non-daily smokers, and recent ex-smokers.

Wave 1			Wave 2									
			Smoking							Not Smoking		
			*Exclusive Smoking*		*Concurrent Use*			Total Smoking				
			Exclusive Daily Smokers (1)	Exclusive Non-Daily Smokers (2)	Dual-Daily Users (3)	Predominant Smokers (4)	Predominant Vapers and Concurrent Non-Daily Users * (5 + 6)	Total Daily Smokers (1 + 3 + 4)	Total Non-Daily Smokers (2 + 5 + 6)	Ex-Smokers Who Vape (7)	Ex-Smokers Not Vaping (8)	Total Discontinued Smoking (7 + 8)
**Exclusive smokers (*n =* 3319)**												
1	Exclusive daily smokers (*n* = 3063)	%	**71.4**	**3.0**	**3.7**	**8.9**	**1.5**	**84.0**	**4.5**	**2.6**	**9.0**	**11.6**
		*n*	*2296*	*79*	*94*	*223*	*37*	*2613*	*116*	*73*	*261*	*334*
2	Exclusive non-daily smokers *(n = 256)*	%	**21.1**	**31.2**	**4.2**	**8.0**	**8.4**	**33.3**	**39.6**	**4.8**	**22.5**	**27.3**
		*n*	*55*	*84*	*7*	*16*	*22*	*78*	*106*	*11*	*61*	*72*
** Concurrent users (*n =* 1117) **												
3	Dual-daily users (*n* = 322)	%	**21.4**	**1.5**	**38.5**	**18.6**	**8.3**	**78.5**	**9.8**	**9.5**	**2.2**	**11.7**
		*n*	*46*	*3*	*129*	*76*	*34*	*251*	*37*	*25*	*9*	*34*
4	Predominant smokers (*n* = 598)	%	**42.2**	**2.7**	**14.9**	**25.4**	**5.4**	**82.5**	**8.1**	**5.0**	**4.5**	**9.5**
		*n*	*216*	*16*	*89*	*187*	*40*	*492*	*56*	*26*	*24*	*50*
5	Predominant vapers (*n* = 88)	%	**4.2**	**1.7**	**12.0**	**18.7**	**39.7**	**34.9**	**41.4**	**18.0**	**5.8**	**23.8**
		*n*	*6*	*3*	*17*	*12*	*29*	*35*	*32*	*15*	*6*	*21*
6	Concurrent non-daily users (*n* = 109)	%	**5.3**	**9.9**	**7.1**	**9.0**	**51.2**	**21.4**	**61.1**	**8.2**	**9.4**	**17.6**
		*n*	*6*	*8*	*10*	*15*	*48*	*31*	*56*	*10*	*12*	*22*
**Recent ex-smokers ^†^ (*n* = 580)**												
			**Relapsed (exclusive smoking or concurrent use)**							**Continued smoking abstinence**		
7	Ex-smokers who vape (*n* = 136)	%	**4.4** (1 + 2)		**8.1** (3 − 6)			**12.5** (1 − 6)		**67.3**	**20.2**	**87.5**
		*n*	*2*		*8*			*10*		*94*	*32*	*126*
8	Ex-smokers not vaping (*n* = 444)	%	**11.3** (1 + 2)		**2.0** (3 − 6)			**13.3** (1 − 6)		**4.6**	**82.1**	**86.7**
		*n*	*47*		*8*			*55*		*21*	*368*	*389*
**Row Totals**												
All daily smokers (*n* = 3983)		%	**67.2**	**2.9**	**5.8**	**10.6**	**2.1**	**83.6**	**5.0**	**2.9**	**8.4**	**11.3**
All non-daily smokers (*n =* 453)		%	**17.7**	**26.1**	**5.5**	**9.2**	**15.5**	**32.4**	**41.6**	**6.5**	**19.6**	**26.1**
All recent ex-smokers (*n* = 580)		%	**10.5** (1 + 2)		**2.8** (3 − 6)			**13.3** (1 − 6)		**12.2**	**74.6**	**86.8**

Data are weighted and adjusted with covariates. The numbering system (1–8) corresponds to Table 1. * Groups 5 and 6 were merged at Wave 2 due to small sample sizes; ^†^ Recent ex-smokers could not be further subdivided at Wave 2 due to small sample sizes. Bold: Weighted estimates are bolded; Italics: sample sizes are italicized.

**Table 4 ijerph-17-07084-t004:** Wave 1 to Wave 2 transition subgroup comparisons: progression towards discontinuing smoking among daily and non-daily smokers and relapse among ex-smokers based on baseline frequency of smoking and vaping.

Wave 1 to Wave 2 Transitions	Transition Code *	Comparisons between Wave 1 Subgroups		OR	95% CI
No change in smoking frequency (daily)	A (i)	1	Exclusive daily smokers	Reference	
		3 vs. 1	Dual-daily users	0.72	0.49–1.05
		4 vs. 1	Predominant smokers	0.91	0.67–1.24
No change in smoking frequency (non-daily)	A (ii)	2	Exclusive non-daily smokers	Reference	
		5 vs. 2	Predominant vapers	0.99	0.51–1.90
		6 vs. 2	Concurrent non-daily users	**2.60**	1.45–4.67
Decreased smoking from daily to non-daily	B	1	Exclusive daily smokers	Reference	
		3 vs. 1	Dual-daily users	**2.41**	1.42–4.10
		4 vs. 1	Predominant smokers	**1.90**	1.23–2.95
Increased smoking from non-daily to daily	C	2	Exclusive non-daily smokers	Reference	
		5 vs. 2	Predominant vapers	1.12	0.58–2.15
		6 vs. 2	Concurrent non-daily users	0.54	0.28–1.04
Discontinued smoking	D (i)	1 + 3 + 4	Daily smokers	Reference	
		2 + 5 + 6 vs. 1 + 3 + 4	Non-daily smokers	**2.48**	1.80–3.43
	D (ii)	1	Exclusive daily smokers	Reference	
		3 vs. 1	Dual-daily users	0.87	0.53–1.41
		4 vs. 1	Predominant smokers	0.75	0.50–1.12
	D (iii)	2	Exclusive non-daily smokers	Reference	
		5 vs. 2	Predominant vapers	0.89	0.43–1.83
		6 vs. 2	Concurrent non-daily users	0.55	0.29–1.06
	D (iv)	3	Dual-daily users	Reference	
		4 vs. 3	Predominant smokers	0.83	0.48–1.45
		5 vs. 3	Predominant vapers	**2.40**	1.07–5.35
		6 vs. 3	Concurrent non-daily users	1.86	0.92–3.75
Relapsed back to smoking	E	7	Ex-smokers who vape	Reference	
		8 vs. 7	Ex-smokers who don’t vape	1.04	0.36–3.03
Initiated Vaping	F	1	Exclusive daily smokers	Reference	
		2 vs. 1	Exclusive non-daily smokers	**1.52**	1.02–2.25
Discontinued smoking and switched to vaping	G (i)	1 + 3 + 4	Daily smokers	Reference	
		2 + 5 + 6 vs. 1 + 3 + 4	Non-daily smokers	**2.21**	1.25–3.91
	G (ii)	1	Exclusive daily smokers	Reference	
		3 vs. 1	Dual-daily users	**3.55**	1.90–6.64
		4 vs. 1	Predominant smokers	**1.88**	1.09–3.23
	G (iii)	2	Exclusive non-daily smokers	Reference	
		5 vs. 2	Predominant vapers	**4.99**	1.82–13.72
		6 vs. 2	Concurrent non-daily users	1.81	0.61–5.35
	G (iv)	3	Dual-daily users	Reference	
		4 vs. 3	Predominant smokers	0.52	0.27–1.01
		5 vs. 3	Predominant vapers	2.25	0.89–5.70
		6 vs. 3	Concurrent non-daily users	1.09	0.44–2.72

Data are weighted and adjusted with covariates. The “group comparison” numbering system refers to Table 1. * Transition codes refer to the group comparisons as described for the second analyses in the Statistical Analyses Section 2.3. Control groups were exclusive smokers (compared to concurrent users). Among concurrent users, dual-daily users were used as the control group (the group with the most frequent smoking and vaping); vs: versus. Bold odds ratios indicate significance. OR: Odds ratio; CI: Confidence interval.

**Table 5 ijerph-17-07084-t005:** Country comparisons among daily smokers and non-daily smokers (past-daily smokers) and reduced smoking and discontinued smoking at Wave 2 (2018) between those who were vaping and not vaping at Wave 1 (2016).

W1 Smoking Status	W1 Vaping	W2 Smoking Status	Canada (*n* = 1411)			United States (*n =* 900)			England (*n =* 1399)			Australia (*n =* 726)		
			*n*	%	95% CI	*n*	%	95% CI	*n*	%	95% CI	*n*	%	95% CI
Daily *n* = 3983	Yes	Remained daily	288	81.0	75.2–85.7	134	66.1	56.3–74.7	255	82.8	76.3–87.8	66	83.0	67.9–91.9
	No		731	82.7	79.5–85.4	519	76.7	70.4–82.0	834	88.4	85.3–90.9	529	85.5	81.1–89.0
	Yes	Reduced to non-daily	43	12.6 *	8.7–17.9	23	16.2^±^	9.9–25.5	21	7.5 ^†^	4.4–12.6	6	6.5	1.3–27.5
	No		37	4.9	3.4–7.0	28	8.4	5.2–13.5	33	2.9	1.7–4.8	18	3.2	1.7–5.9
	Yes	Discontinued smoking	24	6.4 *	3.9–10.3	24	17.7	11.3–26.5	26	9.7	6.0–15.4	10	10.5	4.8–21.6
	No		104	12.4	10.1–15.2	68	14.9	10.5–20.7	99	8.8	6.6–11.5	63	11.3	8.2–15.5
Non-daily *n* = 453	Yes	Remained smoking	61	76.8	64.7–85.7	39	75.5	52.8–89.4	47	86.6	72.0–94.2	7	98.7	88.6–99.9
	No		73	77.1	65.7–78.9	37	54.5	34.6–73.0	57	80.1	66.6–89.0	17	60.2	38.7–78.4
	Yes	Discontinued smoking	23	23.2	14.3–35.3	9	24.5	10.6–47.2	10	13.4	5.8–28.0	1	1.3 *	0.1–11.4
	No		27	22.9	15.1–33.0	19	45.5	27.0–65.4	17	19.9	11.0–33.4	9	39.8 *	21.6–61.3

* *p* < 0.001; ^†^
*p* < 0.05; ±*p* < 0.1. The analyses adjusted for age, education, TIS, and uptake of NVP use between W1 and W2 (regardless of vaping history). Caution is warranted in interpreting any differences within or between countries due to small sample sizes in some cases.

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
