# Peer review of "Changes in Smoking and Vaping over 18 Months among Smokers and Recent Ex-Smokers: Longitudinal Findings from the 2016 and 2018 ITC Four Country Smoking and Vaping Surveys"

_ijerph, 2020, doi:10.3390/ijerph17197084_

Round 1

Reviewer 1 Report

“Transitions in smoking and vaping patterns among exclusive smokers, dual users,
and recent ex-smokers: Longitudinal findings from the 2016 and 2018
ITC Four Country Smoking and Vaping Surveys”

Comments on the paper by S. Gravely et al submitted to the International Journal of Environmental Research and Public Health (IJERPH 882442)

________________________________________________________________

                                                      Date       :    23rd July 2020

While the results of this study are of interest, the presentation of the methods and results needs a comprehensive overhaul before it can be considered publishable.

The abstract is particularly dense and difficult to read.  Lines 25-31 could be much shortened.  For example, “At each wave, individuals were divided into eight groups; exclusive smokers (subdivided as daily or non-daily), dual users (subdivided by each combination of daily/non-daily smoking and daily/non-daily vaping) and recent ex-smokers (subdivided by vaping or not).  Transitions between groups were derived and changes were classified into six groups.”  One of the sentences starting in lines 31 and 32 could be eliminated as saying the same things.  There is perhaps no need to give so many percentages and p values, it being clearer to give the conclusions, and the conclusions could be given more succinctly.  For example, statements like “Vaping at Wave 1 was associated with a significantly (p<0.05) increased likelihood of reducing cigarette consumption from daily to non-daily, but was not significantly related to increasing cigarette consumption, discontinuing smoking or relapsing to smoking.”  Should one say something about overall differences between Wave 1 and 2?  Did smoking decrease?  Did vaping increase?  I didn’t find this in the paper.

Looking at the first section of the methods section, it is stated that the sample recruited had to be cigarette smokers, recent ex-smokers and at least weekly vapers.  The use of the word “and” implies that only dual users were of interest, which is clearly not the case.  The remaining section cites various numbers and percentages which should be in the results section (the same is true for section 2.3).  What is needed is to say what the inclusion criteria were – followed up at W2, had complete data and not never smokers, less than monthly smokers, never daily smokers or quitters for more than two years.  Giving numbers actually adds to the confusion as it is stated that the 5,016 respondents were divided into three groups of size 3,983, 1,117 and 586 which add to 5,580!  I don’t see how “Country NVP Regulations” come under the heading “Study Sample and Procedure”.

Classifying smoking and vaping status in a section headed “Wave 1 baseline measures” is also confusing as smoking and vaping status was clearly determined at both Waves.  Also the text starting on line 141 defining the groups is extremely difficult to follow – how can group 2 be exclusive non-daily smokers and dual users? – and would be better presented in the text in the following style.

At both Waves, individuals were classified into eight groups as follows:

Smoking

Vaping

Exclusive smokers

1.

Exclusive daily smokers

Daily

None

2.

Exclusive non-daily smokers

Non-daily

None

Dual users

3.

Dual daily users

Daily

Daily

4.

Predominant smokers

Daily

Non-daily

5.

Predominant vapers

Non-daily

Daily

6.

Dual non-daily users*

Non-daily

Non-daily

Exsmokers

7.

Ex-smokers who vape

Ex

Daily/Non-daily

8.

Ex-smokers not vaping

Ex

None

(*  Seems a more consistent term than “non-daily concurrent smokers”)

This numbering should be used throughout.  It cannot be right to use a system based on numbers at Wave 1 (as used in the difficult to understand text in lines 143-149), and on letters at Wave 2 (as in Table 2).

I note that section 2.2.2 states that “Transition outcome for smoking and vaping were determined according to the respondent’s  status at W2 ….”.  Surely one needs both W1 and W2 data to determine transitions!  I don’t understand the relevance of the final clause of this sentence “thus one considered point prevalence estimates only.”

A fundamental weakness of the methods section is that no formal statement is made of the hypotheses that are being tested.  This should be made absolutely transparent.  Thus for each test one should clearly present – best in a table perhaps – the change of interest and the groups being compared. 

The classification of transitions into six groups described in section 2.2.2 is difficult to understand because of use of terms such as “cigarettes/NVPs” or “smoking/vaping”, when it is unclear whether the “/” implies “and”, “or”, or “and/or”. 

As regards presentation of the results it would be useful to have a single table giving all the transitions, with a row for each of the eight Wave 1 groups and a column corresponding to each of the eight Wave 2 groups, plus marginal totals in both directions.  For each cell, one should give the numbers of individuals the weighted % of the group 1 value, and perhaps which of the six change groups defined in section 2.2.2 the cell related to.  It seems bizarre to define the six groups, and then apparently not use them.  Total numbers in each of the six change-groups could be given in the footnote perhaps.  The rows in Table 2 for smoking and vaping are not necessary.  Table 3 states that the data are weighted and adjusted.  Is this the same thing?  This is not clear.

The text describing the tables should try to discuss the results in general terms and not give lots of percentages that are clearly in the Table.

Table 5 is also quite difficult to follow, and seems to include repetitions.  Why, for example, compare exclusive daily smokers, predominant smokers and dual daily users at Wave 1 for whether they are still smoking and for whether they have discontinued smoking?  One test is enough here.  (I note the same weaknesses in the abstract.)  Given all the transition rates are in the transition table why not just give the odds ratios and 95% CIs (which is what I assume these undescribed values are).  Perhaps the Table would be easier to understand if it were recast something like:

Change

Switching from daily to non-daily

Exclusive daily smokers

Reference

smoking

Predominant smokers

0.91 (0.67-1.24)

Dual daily users

0.72 (0.49-1.05)

Quitting daily smoking

Exclusive daily smokers

Reference

Predominant smokers

0.75 (0.50-1.12)

Dual daily users

0.87(0.53-1.41)

Quitting daily smoking and

Exclusive daily smokers

Reference

becoming exclusive vapers

Predominant smokers

1.88 (0.09-3.23)

Dual daily users

3.55 (1.90-6.64)

Enough comments for now, until I have seen a clearly laid out and presented revision!

Except that I noted that in the introduction there is a totally unjustified statement that “It is now well established that NVPs are less harmful to smokers than cigarettes.”  How can this possibly be true, given the lack of good epidemiological data showing that smokers switching to e-cigarettes actually do have reduced risks of disease compared to smokers who continue to smoke, or that long term exclusive e-cigarettes users actually do better than exclusive smokers.  There may be good arguments that risks ought to be lower, but that does not justify use of the word “well-established”.

Author Response

Manuscript ID: ijerph-882442: Transitions in smoking and vaping patterns among exclusive smokers, dual users, and recent ex-smokers: Longitudinal findings from the 2016 and 2018 ITC Four Country Smoking and Vaping Surveys

Please find responses below to the peer-review of this paper. Please note that the page numbers herein refer to the manuscript with tracked changes.

We sincerely thank the reviewers for their time and insightful feedback for this paper.

Reviewer 2 Report

The paper analyzes two surveys taken 18 months apart in a longitudinal study of nicotine use in 4 English-speaking countries, Waves 1 (2016) and 2 (2018) of the ITC 23 Four Country Smoking and Vaping Survey, to describe the evolution of nicotine use and sources amongst current and former smokers as a function of Nicotine Vaping Product (NVP) use. The paper mainly addresses the transitions within a 3x3 matrix (smoking daily, less, never x vaping daily, less, never) of 5,000 respondents from wave 1 to wave 2. It does not address vaping never-smokers (potential “gateway” effects of NVP). The paper secondarily looked for consequences of policy differences, as England endorses NVP as a harm reduction strategy, the USA permits but does not endorse NVP, Canada prohibits NVP without much enforcement, and Australia prohibits NVP with strong enforcement. Vaping was associated with reduced daily smoking over the 18 month period, but not with increased abstinence from smoking. Aggressive regulation was associated with small sample sizes for NVP users and big p values.

Critique

The paper is quite important as a broad description of the natural history of cigarette and NVP use in the recent past. Overall, the paper is refreshingly even handed and restrained in that it uses the language of association rather than causation, acknowledges potential benefits of NVP, briefly acknowledges in the discussion the varied clinical scenarios in which NVP are used, and recognizes that even with NVP displacing some smoking, the persistence of nicotine use patterns, especially smoking, over 18 month periods is frustratingly high from a public health perspective.

The paper is also ambitious and the results tedious as it describes possible transitions from most of the 9 cells in the first 3x3 matrix to each of the 9 cells in the second 3x3 matrix. It is a daunting task to make this information engaging verbally or visually. The current manuscript provides detail and precision, but the results and tables are difficult to assimilate. I think tables 2, 3, and perhaps 5 could be illustrated with visually meaningful horizontal bar charts that would complement, not replace, the tables. My best off hand suggestion for these charts would be as follows:

  • Make a row for each W1 subdivision for the category represented in the table, with the height of the row proportional to N or log(N) in the subdivision. Use a column of constant width cells to represent the W1 subdivision, and color code and mark each cell to match the corresponding subdivision as in the next step.

  • Make a horizontal bar for the W1 category divided by color code and heavy black lines into three main categories representing the main W2 classification (Daily Smokers, Non-daily Smokers, and Discontinued Smoking for tables 2 & 3), with the width of the bar proportional to the wave 2 weighted percentage. Color code the worse situations with darker colors, so the bar gets lighter from left to right. All bars have the same width, representing 100%, but different areas due to the height, so that area represents N or log(N).

  • Use thin black lines to subdivide the three W2 classifications in each row as in the table, e.g. Daily Smokers is subdivided into Exclusive smokers, Predominant smokers, and Dual users. Make these widths proportional to %. Mark each subdivision either with a key word or using V’s to suggest the proportion of nicotine taken by Vaping – sparse V’s for Predominant smoking, more dense V’s for Dual use. The V’s could be different colors to contrast sharply with whatever background color marks the category (e.g. light V’s on the left and dark V’s on the right).

Tables 2 and 3 could be combined into one illustration without being overwhelming, I think. This illustration should visually make an emphatic point that the non-daily smoking expands more at W2 for people who were using some NVP at W1, but that discontinued smoking is equally elusive for all.

Page 15, line 10 (because the numbering restarts after tables 4, 5, and 6), “This association was not found in the US (16.2% vs. 8.4%, p=0.063) or in Australia (6.5% vs. 3.2%, p=0.54).” Alas, we are slaves to the p value. Obviously the US trend is in the same direction as Canada and England, and certainly should not be taken as conflicting evidence. The Australian sample is apparently too small to get more than a coin toss p value out of a doubled rate, again in the same direction, and more likely due to a small sample than conflicting evidence. A sequential Bayesian revision incorporating each country in turn would yield quite a skewed distribution of posterior probabilities, I expect. The last sentence in this paragraph could reasonably be reworded to acknowledge that the direction of influence is the same in all four countries, but conventionally statistically significant in two, trending in the third, and indeterminate in the fourth where NVPs were most strongly prohibited. Page 18, lines 58-72 also might deserve some editing in this light – there are no contradictions in the direction of this effect, only in the p-value, and we suspect that we know why.

It seems unfair to ask for added content in the current manuscript, but the survey has more light to shed on this topic, as the authors suggest. 4CV1 question 105 is likely to correlate with nicotine receptor alleles (How easy or hard would it be for you to quit smoking if you wanted to?). Questions 111-115 are likely to correlate with CYP alleles affecting nicotine metabolism rate (numbers of cigarettes/time). Unfortunately, 4CV1 contains no questions about some major risk factors for smoking, especially a history of adverse childhood events, psychiatric illness, and family smoking history. It does have a question about drinking alcohol, which should be helpful. The probabilities that a smoker attempts to quit smoking and that quitting is successful almost certainly are dependent on these genetic and historical features. The authors seem to appreciate these issues, and I hope that they will submit more analyses to explore them.

Another way to look at these smoking scenarios is that daily vaping is a marker for a person who feels unable to break away from nicotine but is worried about the health consequences of smoking: it is not necessarily realistic to imagine that nicotine abstinence is always an attainable goal. Occasional vaping is a marker for a person who vapes when social situations prohibit or encourage smoking. In these scenarios it would not necessarily be appropriate to think of vaping as a smoking cessation intervention. The authors also seem to appreciate these nuances, and I hope that they will submit more analyses along these lines.

Author Response

(The authors gave the same response as above.)

Reviewer 3 Report

This manuscript reports the descriptive findings from two waves of a smoking and vaping survey conducted across four developed countries: Australia, Canada, USA and the UK. This is important descriptive information to provide a basis to exploring the effectiveness of nicotine vapour products in supporting the cessation of cigarette smoking. 

The manuscript is very well written, the results from the analyses are clearly reported in well designed tables and the authors' discussion points, including the strengths and limitations of the study, are supported by the results provided. It is very interesting reading and provides important findings to support further research in this field. Their findings are reported within the contexts of the varying tobacco and nicotine vapour policies of each country, which demonstrates the heterogeneity of approaches between countries in tackling the contribution of tobacco to the health burden. 

I have no concerns about this paper in its current form and recommend it be accepted in its current form. 

Author Response

(The authors gave the same response as above.)

Round 2

Reviewer 1 Report

Compared to the earlier version, the latest version is somewhat improved.  However, further improvement is required.

While the classification of joint smoking and vaping status into eight groups, as given in Table 1, is fine, there is a major problem with the classification of transitions, as this is inconsistently defined and very confusing to the reader:

In the abstract we have seven numbered groups

1 = no change

2 =increased smoking

3 = decreased smoking

4 = discontinued smoking

5 = initiated vaping

6 = switched from smoking to exclusive vaping

7 = relapsed.

In lines 169-173, we have the same seven groups, but what were 5, 6 and 7 in the abstract are now renumbered 6, 7 and 5 for no apparently good reason.  In lines 193-306, we have a different set of seven groups.  Compared to the numbering system in the abstract, 1 is renamed “continued smoking” but 6 and 7 are numbered differently.

In Table 3, we only have three transitions, while in Table 4 we have nine.

This, frankly, is terribly confusing to the reader.  I would recommend the following:

  1. Use the same list of transitions throughout, and name them consistently.
  2. Reference the transitions by letters A, B, C ……, using letters to avoid any conflict with the group numbers and refer to those codes in Tables of results. If some of the transitions are subsets of a broader transition, one could indicate this in the coding system.  Thus A = no change could be subdivided into A1 = continue to smoke only, A2 = continue to smoke non-daily, A3 = continue to be ex-smokers.
  3. In defining the transitions it would be useful to have a table similar to the current Table 3, but giving all the transitions with separate transitions.
  4. Do not, in section 2.3 statistical analyses, attempt to redefine the transitions. Simply say that the analyses will relate the frequency of transitions A1, A2, B, C, etc. to which group the individuals were at W1.

One problem with the paper is that it gives no clear indication of how habits changed between W1 and W2.  Has smoking declined?  Has vaping increased?  A simple solution to this problem might be to extend Table 1 so that it additionally gives the breakdown of the population into the eight groups separately for Wave 1 and Wave 2.  Table 1 could then appear at the start of the results section rather than in section 2.2.3, possibly after the existing Table 2 on baseline characteristics.  A full description of the transitions and their frequency is given in the results section, so it seems logical to give a full description of the groups and their frequency at each Wave also in the results section.

The first sentence of the abstract needs rewriting.  How can you define smoking patterns based on baseline vaping status between two time points?  Should not the title more simply be “Changes in smoking and vaping over an 18 month period in smokers and recent ex-smokers in the ITC Four Country Smoking and Vaping Surveys”?  It is much too convoluted as it stands.  Say in general terms what you are doing, define the groups, define the transitions and say that the analyses relate transition rates to baseline smoking/vaping habits. 

A few other points going through the paper in order are as follows:

Line 35         For transition 7 one could omit “who were vaping or not vaping” and just say “relapsed: ex-smokers at W1 who were smoking at W2”.

Lines 36-43       In various places sentences have the construct “Group 1 was more likely to do A and B than group 2”.  The sentences would read better if the “than group 2” were to go after the “likely”.

Lines 156-157     The sentence starting “Concurrent users …..” might better start “The categorization of concurrent users was according …..”.

Line 169     “focused only on smoking” might be better.

Lines 228-247     Why do the percentages not add up to 100%?  For “W1 exclusive daily smokers”, the numbers in Table 3 indicate the correct figures are as follows:

Transition

W2 Group

%

Did not change smoking (were still exclusive daily smokers)

1

71.4

84.0

Did not change smoking (and took up vaping)

3 + 4

12.6

Reduced smoking (remained non-vapers)

2

3.0

4.5

Reduced smoking (and took up vaping)

5 + 6

1.5

Quit smoking (regardless of vaping)

7 + 8

11.6

11.6

100.1

Thus the missing bit in your text is the 12.6%.  Clearly all percentage distributions must add to 100% unless it is clearly explained why.

Table 3 should give the eight groups first and then any combinations afterwards.

Table 4 results page 10.  Same problem as previously described, with the “than” being at the end of the sentence when it would be better than earlier.

Table 4.  Given that it is clearly stated what the reference group is, there is no need to have the “vs” in the group column.  Put the transition codes in the table.

Why in Table 4 are those who are daily smokers and daily vapers called dual daily users, and those who are non-daily smokers and non-daily users called concurrent non-daily users?  Inasmuch as those who are non-daily users of both might smoke on Tuesdays and vape on Wednesdays, how does concurrence come into it?

Discussion generally.  In my view all sentences ending “compared to” and “than” would read more easily if the comparator is placed together with the group with which is it being compared.

Lines 19-21 in discussion.  I do not understand this.  How can smokers who do not stop smoking be more likely to stop?

I found the final sentence of the conclusions very difficult to take in.  The first part might be written more comprehensibly something like “Smokers who vaped were more likely than exclusive smokers to reduce their smoking, but were not more likely to quit smoking”.  Can we not mention relapse in the conclusions?

Author Response

*see attached document

Round 3

Reviewer 1 Report

“Changes in smoking and vaping over 18 months among smokers and recent ex-smokers:
Longitudinal findings from the 2016 and 2018 ITC
Four Country Smoking and Vaping Surveys”

Comments on the second revision of the paper by S. Gravely et al submitted to
the International Journal of Environmental Research
and Public Health (IJERPH 882442)

_______________________________________________________

We are getting much closer to a very good paper!  I thank the authors for their detailed response to my earlier comments.  However I still have a few points as detailed below in the order they appear in the paper.

Lines 22-24.   The first sentence of the abstract is still unclear.  I reiterate my point that you cannot classify changes in smoking and vaping based only on baseline vaping.  Also the 18 months apart is too far removed from the “changes in”.  I suggest something like: “This study of smokers (smoked at least monthly) and recent ex-smokers (quit for < 2 years) relates changes over an 18 month period in their smoking and vaping habits to their smoking and vaping status at baseline.”

Line 94.  As you were discussing the PATH study in the previous paragraph, starting “This study” sounds as if you still are.  Better to start something like”:  “Here, we use data from the longitudinal ITC Four Country and Vaping Study (ITC 4CV), to explore and describe…” 

Line 111.   Given the previous change, this could simply start “ITC 4CV” is a longitudinal cohort study….”

Lines 150-155.  There is no need to describe the groups both in the text and Table 1.  Just say “The respondents were categorized in terms of daily and non-daily smoking and vaping as described in Table 1.  The categorization of concurrent users is according to the system developed by Borland et. al. [22].”

Lines 170 and 171.  These should be separated by a blank line.

Lines 180-182.  “.. and descriptively examined..”  should be “.. and to descriptively examine” to fit in with the earlier “to estimate”.

Line 258 (Table 3).  Although the numbers in Table 3 add up OK, I think there are some errors in the colouring.  Thus those who are Wave 1 group 3 and Wave 2 group 4 are daily smokers at both time points so should be blue not red.  Those who are Wave 3 groups 5 and 6 and Wave 2 group 2 are both non-daily smokers at both time points so should also both be blue not red.  The rest are OK I think, though this might benefit from checking.

Lines 266 to 296.  One might consider whether it is actually necessary to give all the ORs and 95%CIs in the text, since they are all in Table 4.  The headings might also be put more into a readily understandable form.  I won’t attempt to rewrite the whole section but I will have a try at a suggestion for the first section.

“Comparisons of Wave 1 Daily and Non-Daily Smokers

The results for transitions D (i), F and G (i) show that, compared to daily smokers, non-daily smokers were significantly (p<0.05) more likely  to discontinue smoking, to initiate vaping, and to do both.”

Line 299 (Table 4).  The heading “Groups Comparisons (Status at Wave 1)” might better read “Comparisons made between Wave 1 Groups”.

Discussion line 33.  Please add a comma after “6 months”.

Discussion lines 64 and 65.  This sentence might better read “Disconcertingly, among daily smokers in Canada, discontinuation of smoking was less likely in those who vaped than in those who did not vape.”

Author Response

Responses to Reviewer 3: Round 3

  1. We are getting much closer to a very good paper!  I thank the authors for their detailed response to my earlier comments.  However I still have a few points as detailed below in the order they appear in the paper.

Authors’ Response: We again sincerely thank the reviewer for their time and effort in reviewing this paper. It has indeed helped shape this paper into a great manuscript. We have therefore added an acknowledgement at the end of the manuscript to show our appreciation.

  1. Lines 22-24.   The first sentence of the abstract is still unclear.  I reiterate my point that you cannot classify changes in smoking and vaping based only on baseline vaping.  Also the 18 months apart is too far removed from the “changes in”.  I suggest something like: “This study of smokers (smoked at least monthly) and recent ex-smokers (quit for < 2 years) relates changes over an 18 month period in their smoking and vaping habits to their smoking and vaping status at baseline.”

Authors’ Response: Due to limited space in the abstract, we have used a partial reiteration of the above. The sentence now reads as:

Line 22: “This descriptive study of smokers (smoked at least monthly) and recent ex-smokers (quit for ≤ 2 years) examined transitions over an 18 month period in their smoking and vaping behaviors”.

  1. Line 94.  As you were discussing the PATH study in the previous paragraph, starting “This study” sounds as if you still are.  Better to start something like”:  “Here, we use data from the longitudinal ITC Four Country and Vaping Study (ITC 4CV), to explore and describe…” 

 Authors’ Response: This is an excellent point. This sentence has been revised:

Line 94: “In this current prospective cohort study, data from the ITC Four Country and Vaping Surveys (ITC 4CV) were used to explore and describe…”

  1. Line 111.   Given the previous change, this could simply start “ITC 4CV” is a longitudinal cohort study….”

 Authors’ Response: We have made this edit (now line 112).

  1. Lines 150-155.  There is no need to describe the groups both in the text and Table 1.  Just say “The respondents were categorized in terms of daily and non-daily smoking and vaping as described in Table 1.  The categorization of concurrent users is according to the system developed by Borland et. al. [22].”

Authors’ Response: We have made this change.

  1. Lines 170 and 171.  These should be separated by a blank line.

Authors’ Response: Thank-you for noticing this (which must have occurred on the editor’s side after accepting the tracked changes). We have added a blank line in this location (and formatted in other places where needed).

  1. Lines 180-182.  “.. and descriptively examined..”  should be “.. and to descriptively examine” to fit in with the earlier “to estimate”.

Authors’ Response: We agree that this could be more concise. We have revised this as:

“First, multivariable logistic regression models were used to descriptively examine within-person transitions between states of smoking and vaping (based on Table 1 group classifications: 1-8)…”

  1. Line 258 (Table 3).  Although the numbers in Table 3 add up OK, I think there are some errors in the colouring.  Thus those who are Wave 1 group 3 and Wave 2 group 4 are daily smokers at both time points so should be blue not red.  Those who are Wave 3 groups 5 and 6 and Wave 2 group 2 are both non-daily smokers at both time points so should also both be blue not red.  The rest are OK I think, though this might benefit from checking.

Authors’ Response: We thank the reviewer for noting the colour errors. We have had to removed the colour coding from the table as per IJERPH’s journal guidelines.

  1. Lines 266 to 296.  One might consider whether it is actually necessary to give all the ORs and 95%CIs in the text, since they are all in Table 4.  The headings might also be put more into a readily understandable form.  I won’t attempt to rewrite the whole section but I will have a try at a suggestion for the first section.

“Comparisons of Wave 1 Daily and Non-Daily Smokers

The results for transitions D (i), F and G (i) show that, compared to daily smokers, non-daily smokers were significantly (p<0.05) more likely  to discontinue smoking, to initiate vaping, and to do both.”

Authors’ Response: We have deleted the ORs and 95% CIs in the text on page 10 (referring to Table 4). In their place, we have added the transition code. We have also updated (and simplified the sub-headings) on page 10. For example:

Table 4 presents the subgroup comparisons (including the transition code) for each of the seven transition outcomes. The main differences are outlined below.

Comparisons Between Wave 1 Daily and Non-Daily Smokers

Non-daily smokers were more likely than daily smokers to have discontinued smoking [transition code: D (i)] or to have switched to vaping [G (i)]. Exclusive non-daily smokers were more likely than exclusive daily smokers to have initiated vaping between W1 and W2 (F).

  1. Line 299 (Table 4).  The heading “Groups Comparisons (Status at Wave 1)” might better read “Comparisons made between Wave 1 Groups”.

Authors’ Response: We have changed the table sub-heading to: ‘Comparisons between Wave 1 Subgroups’

  1. Discussion line 33.  Please add a comma after “6 months”.

Authors’ Response: We have added the comma.

  1. Discussion lines 64 and 65.  This sentence might better read “Disconcertingly, among daily smokers in Canada, discontinuation of smoking was less likely in those who vaped than in those who did not vape.”

Authors’ Response: We have made this change.